# Impact Analysis of Univariate and Multivariate Bias Correction on Rice Irrigation Water Needs in Jiangxi Province, China

**Wido Hanggoro [1,2], Jing Yuanshu [1,*], Leila Cudemus [1,3] and Jing Zhihao [1]**

[1] Collaborative Innovation Center on Forecast and Evaluation Meteorological Disasters/Jiangsu Key Laboratory of Agriculture Meteorology, Nanjing University of Information Science and Technology, Nanjing 210044, China; wido_hanggoro@yahoo.com (W.H.); leilacudemus@hotmail.com (L.C.); jingzhhao@163.com (J.Z.)

[2] Research and Development Center of Indonesia Agency for Meteorology Climatology and Geophysics, Jakarta 10720, Indonesia

[3] National Institute of Meteorology and Hydrology of Venezuela, Miranda 1080, Venezuela

* Correspondence: appmet@nuist.edu.cn; Tel.: +86-25-58699815

**Abstract:** Regional climate models (RCMs) provide an improved representation of climate information as compared to global climate models (GCMs). However, in climate-agricultural impact studies, accurate and interdependent local-scale climate variables are preferable, but both RCMs and GCMs are still subjected to bias. This study compares univariate bias correction (UBC) and multivariate bias correction (MBC) method to simulate rice irrigation water needs (IWNs) in Jiangxi Province, China. This research uses the daily output of Hadley Centre Global Environmental Model version 3 regional climate model (HadGEM3-RA) forced with ERAINT (ECMWF ERA Interim) data and 13 Jiangxi ground-based observations, and the observation data are reference data with 1989–2005 defined as a calibration period and 2006–2007 as a validation period. The result shows that UBC and MBC methods favorably bias-corrected all climate variables during the calibration period, but still no significant difference is noted between the two methods. However, the UBC ignores the relationship between climate variables, while MBC preserves the climate variables' interdependence which affect subsequent analyses. In rice IWNs simulation analysis, MBC has better skill at correcting bias compare to UBC in $ET_o$ (evapotranspiration) and $P_{eff}$ (effective rainfall) components. Nonetheless, both methods have a low ability to correct extreme values bias. Overall, both techniques successfully reduce bias, even though they are still less effective for precipitation compared to maximum and minimum temperature, relative humidity and windspeed.

**Keywords:** univariate; multivariate; bias correction; irrigation water needs; regional climate model; Jiangxi Province

## 1. Introduction

Global climate models (GCMs) have been widely used to assess climate effects on nature and human society [1]. They are designed to simulate the climate system and to also effectively address large-scale climate features such as general circulation of the atmosphere and the ocean, and sub-continental patterns of, for example, temperature, precipitation, and many physical processes in [2–4]. Nonetheless, GCMs provide inadequate information on regional-to-local scale due to poor resolution and incomplete representation of physical processes [5–8].

Regional climate models (RCMs) are one valid approach to derive climate information at a higher resolution (typically 10–50 km) through dynamical downscaling driven by lateral boundary conditions from GCM [9–12]. This resolution offers a more detailed representation of topography and

many dynamical and physical processes which is better for simulating climate at a regional and local scale [13–15].

However, RCMs are still subjected to considerable biases when the simulation is compared to observations, even at the same spatial scale [16]. Bias is generated from both incorrect boundary condition (GCMs' forcing data) and by systematic model error sourced from the flawed equations, discretization, and model grid averaging [6,17–19]. As a result, raw climate model data cannot be directly used in climate impact studies due to the presence of biases in the representation of regional climate and, therefore, first need to be bias-corrected [20,21].

Several bias-correction methods which use univariate approaches have been developed and used in previous studies [5,9,22–27]. The univariate techniques correct individual variables separately and independently from one another, which can be problematic especially in cases involving multiple, interrelated variables. This neglected relationship or dependence that exists between climate variables thus generates unrealistic results that might degrade the validity of downscaling and impact studies [28–32]. Multivariate bias-correction approaches have been developed to remove biases while still conserving the physical relationship between variables represented by the observation data as a reference [31].

In the agricultural-climate impact context, irrigation water needs (IWNs), which are dependent on local climate conditions, are one of the essential and challenging factors (involving multiple climate variables) that need to be assessed. Thus, it is a crucial step to further adjust the output of RCMs to agree with local climate conditions at specific meteorological stations especially in a region with large topographic variabilities [33]. Many previous studies have focus on the analysis of station-scale bias adjustment on hydrological system [19,34], but only a few of them concentrated on the IWNs (e.g., Smith et al. [33] and Rasmussen et al. [35] who used the univariate approach). Moreover, only a few studies compare the performance of univariate and multivariate methods. As these comparative studies still focus on a regional scale [36,37], studies comparing univariate and multivariate bias correction methods at the station are still scarce, especially those considering many climate variables.

In this study, we focus on southeast China's Jiangxi Province, which is an important agricultural region with abundant natural resources [38]. It is characterized by complex topography and land cover comprising of approximately 36% mountainous area, 42% hills, and 22% mounds, plains, and water bodies [39]. Jiangxi's climate is mainly influenced by the East Asian monsoon with the most distinctive feature in the annual rain cycle being the Meiyu front (intense precipitation cloud band) [40]. Moreover, Jiangxi also experiences the strongest El Nino Southern Oscilation (ENSO)-related climate signals in China [41]. With these climate driving factors and topography uniqueness, climate information varies greatly from one region to another within this province. Therefore, accurate local climate information is needed to improve the calculation of IWNs from regional-scale climate models. Thus, the present study evaluates the performance of station-scale univariate bias correction (UBC) and multivariate bias correction (MBC) on rice IWNs in Jiangxi Province. The remaining sections are organized as follows: Section 2 presents the study area and data, and Section 3 introduces the methods, while results and discussions are enumerated in Section 4. Finally, conclusion and possible recommendations are presented in Section 5.

## 2. Study Area and Data

### 2.1. Study Area

Jiangxi Province is located in the red soil low-hilly area of Southeast China (Figure 1). It has a subtropical monsoon-type climate with wet summers and dry autumns [42]. It has an annual average temperature of 11.6–19.6°C and yearly precipitation of 1637.9 mm and a recorded precipitation maximum in June and minimum in December [41,42]. However, the seasonal distribution is uneven on a time basis, and the rainy season (April to June) accounts for about 50% of the annual precipitation [43].

Agriculture is one of the leading sectors in Jiangxi province. It is the third largest rice-producing province, accounting for 11% of China's output, and is one of the two largest suppliers of rice in

China [41]. It has a yearly average of double-rice yield ranging from 4825.54 to 5608.35 kg/ha during the 1980–2012 period [44].

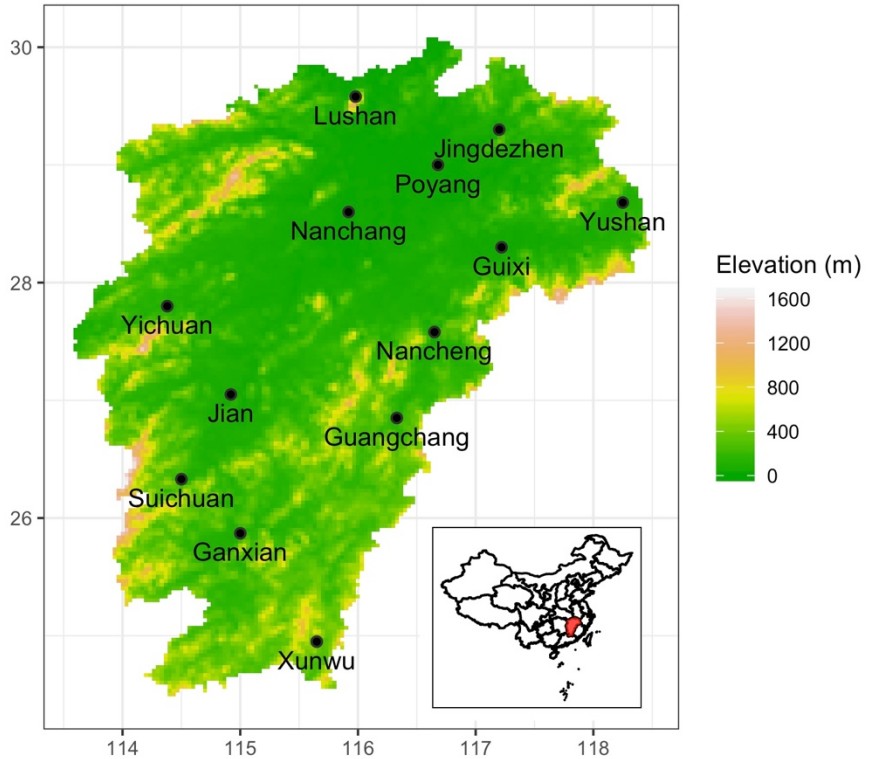

**Figure 1.** Map of Jiangxi Province showing the location of the ground-based observation stations; Inset map shows the location of Jiangxi Province (marked red) within China.

*2.2. Data*

2.2.1. Observation

The daily observed meteorological data from 13 different ground-based observations over Jiangxi Province China (Table 1), covering 1989–2007, is obtained from the China Meteorological Administration (CMA). These include; precipitation (Pp), maximum temperature (Tx), minimum temperature (Tn), relative humidity (RH), and wind speed (Ws).

**Table 1.** Detailed information of ground-based observations used in the study.

| Station | Latitude | Longitude | Elevation (m) |
|---|---|---|---|
| Ganxian | 25.52 | 115.00 | 137.5 |
| Guangchang | 26.51 | 116.20 | 143.8 |
| Guixi | 28.19 | 117.15 | 60.8 |
| Ji'an | 27.03 | 114.55 | 71.2 |
| Jingdezhen | 29.18 | 117.12 | 61.5 |
| Lushan | 29.35 | 115.59 | 1164.5 |
| Nanchang | 28.36 | 115.55 | 46.9 |
| Nancheng | 27.35 | 116.39 | 80.8 |
| Poyang | 29.00 | 116.41 | 40.1 |
| Suichuan | 26.20 | 114.30 | 126.1 |
| Xunwu | 24.57 | 115.39 | 303.9 |
| Yichuan | 27.48 | 114.23 | 131.3 |
| Yushan | 28.41 | 118.15 | 116.3 |

### 2.2.2. Regional Climate Model

The regional climate model data (hereafter referred to as RAW) is obtained from the East Asia Coordinated Regional Climate Downscaling (CORDEX-EA) phase I, under the World Climate Research Program (WCRP) project [45]. It is one of the CORDEX-EA participant models from the National Institute of Meteorological Sciences (NIMS) South Korea. The NIMS utilizes HadGEM3-RA, which is very similar to the HadGEM3-A [46,47], forced with the European Centre for Medium-Range Weather Forecasts Re-Analysis (ERAINT) [48]. The model product a has $0.22° \times 0.22°$ resolution (approximately 25 km), with the $396 \times 251$ grid points in the west–east and north–south directions, respectively [49]. The HadGEM3-RA is limited area model which share common atmospheric and land surface model components with HadGEM3. The dynamic core of HadGEM3-RA is a non-hydrostatic, compressible, and deep atmosphere equation, which has ability to simulate weather and climate modeling at very high resolution [46]. Previous research has established that the HADGEM3-RA simulates a realistic monsoon without the benefit of pseudo-observational forcing at lateral boundaries in west Africa [50].

### 2.2.3. Rice Crop Information

The variables used for the calculation of rice IWNs are time- and site-specific. Therefore, direct observation or a survey is required to determine the number of variables that are often difficult to find for each specific location. To overcome this problem, this study collects several parameters from previous related research and applies them. This information is in general form but still relevant (Table 2).

**Table 2.** Crop calendar information during the rice late period (01 June–11 November) in Jiangxi Province for the 2006–2007 simulation period.

| Growth Stages | Duration (days) | Kc | SAT (mm/month) | WL (mm/month) | Perc (mm/day) |
|---|---|---|---|---|---|
| Land Preparation | 30 | - | 200 | - | - |
| Initial | 30 | 1.04 | - | 100 | 5 |
| Developing | 30 | 1.25 | - | - | 5 |
| Middle | 40 | 1.46 | - | - | 5 |
| Late | 30 | 1.03 | - | - | 5 |

Some information, including crop calendar and growth stages, are needed. From the references, it can be concluded that rice, which has a total rice growth duration ranging from 125–150 days [51–53], is planted two or even three times a year in Jiangxi Province [44,54]. According to Berge et al. [55], the first period of preparation for planting to harvest (early rice period) starts around March–July, while the late rice period starts around April–November.

The value of the rice crop coefficient for each stage in Jiangxi Province is based on the research by Liu et al. [56], who examined variations in water demand in Jiangxi Province during 1987–2009. While soil-water information such as saturated water content, water level, and percolation is obtained from the general information contained in the study of Allen et al. [53].

### 3. Methods

The focus of this research is a performance comparison between univariate and multivariate bias correction methods for all five climate variables (Pp, Tx, Tn, RH, Ws), that are related to rice IWNs calculation. Since the model output did not provide relative humidity (rh) as a final product, specific humidity (q) from the model output is converted to rh. The calculation is performed using the Humidity R package [57], which is based on the Clausius–Clapeyron equation [58].

Furthermore, both methodologies are tested for the 1989–2007 simulation period. This study used a split-sample test, which is commonly applied to determine how well bias-correction methods

perform under changing conditions [59]. The calibration period is 1989–2005, while 2006–2007 is used to simulate rice IWNs as a validation period.

*3.1. Univariate Bias Correction (UBC)*

The UBC method establishes the quantile mapping (QM) technique. The QM algorithms are commonly used to correct systematic distributional biases in precipitation [60,61]. This method is based on quantile association and has many names in literature such as statistical downscaling, quantile mapping, histogram equalizing, and rank matching [26,30]. The QM method has been widely used in previous studies and has shown excellent results in correcting model bias [5,6,17,26,27].

This method uses a statistical transformation approach with the assumption that the distribution of the model data is approximated by a stationary correction factor. According to Gudmundsson et al. [27], the general equation is described as follows:

$$P_o = h(P_m) \tag{1}$$

Since statistical transformations are the application of the probability integral function, it can be defined as Equation (2) if the distribution of the variable of interest is known [27].

$$P_o = F_o^{-1}(F_m(P_m)) \tag{2}$$

where $F_m$ is the cumulative density function (CDF) of $P_m$ and $F_o^{-1}$ is the inverse CDF (or quantile function) corresponding to $P_o$. To solve Equation (2), the empirical quantiles approach, which is available in the qmap package on the R software [62], applied. The same method is used by Boé et al. [10] and Themeßl et al. [63].

*3.2. Multivariate Bias Correction (MBC)*

Several studies using the MBC method have been conducted (e.g., Bürger et al. [64], Vrac and Friederichs [30], Mehrotra and Sharma [8], and Cannon [4]). The MBC methods have been used as an improvement to the previously popular UBC methods.

The MBC method in this study is obtained from the MBCn package in the R software [65] developed by Cannon [21]. This package uses the MBCn algorithm, which is a development of the N-dimensional probability density function transfer (N-pdft). The idea of N-pdft is to solve the N-dimensional problem using a succession of one-dimensional distribution transfer problems. It can be done by manipulating N-dimensional to one dimensional probability density function (pdf) using radon transform [66].

The MBCn is a combination of the previous quantile delta mapping (QDM) technique with random orthogonal function to follow the multivariate distribution of model and observation data [31]. The QDM transfer function for quantile mapping that preserves absolute changes in quantiles is given by [4]:

$$\hat{x}_{m,p}(t) = F_{o,h}^{-1}\left\{F_{m,p}\left[x_{m,p}(t)\right]\right\} + x_{m,p}(t) - F_{m,h}^{-1}\left\{F_{m,p}\left[x_{m,p}(t)\right]\right\} \tag{3}$$

where $\hat{x}_{m,p}(t)$ is bias corrected value at time t, $F_{o,h}^{-1}$ and $F_{m,h}^{-1}$ is inverse CDF and $F_{m,p}\left[x_{m,p}(t)\right]$ and $F_{m,p}\left[x_{m,p}(t)\right]$ is CDF function. *o*: observed, *m*: modeled, *h*: historical, and *p*: prediction. The MBCn algorithm is constructed using three main steps: first, conversion of N-dimensional pdf forms to one-dimensional pdf using the N-pdft algorithm; secondly, correct the converted one-dimensional marginal distribution using Equation (3); and, lastly, rotate back the QDM-corrected dataset and convergence to the multivariate distribution is checked. In this study, we apply 30 iterations until the multivariate distributions of bias-corrected modelled and observed climate data match.

### 3.3. Irrigation Water Needs (IWNs)

Water is one of the most vulnerable agriculture elements under climate change besides light and heat conditions [67]. The IWNs explain the water requirements, which make a plant grow in favorable conditions. The basic IWNs formula based on Döll and Siebert [51], and Boonwichai et al. [68] is as follows:

$$IWNs = ET_c - P_{eff} = ET_o * K_c - P_{eff} \text{ if, } ET_c > P_{eff} \quad IWNs = 0 \text{ if, } ET_c \leq P_{eff} \tag{4}$$

where IWNs are partial water needs (a combination of rainwater and irrigation) for a crop, $ET_c$ is crop evapotranspiration, and $P_{eff}$ is an effective rainfall. Effective rainfall is part of total precipitation on the cropped area, during a specific period, which is available to meet evapotranspiration in the cropped area [69]. The effective rainfall equation is taken from the U.S. Department of Agriculture Soil Conservation Method approximation as cited by Smith [70]:

$$P_{eff} = P(4.17 - 0.2P)/4.17 \text{ for, } P < 8.3 \text{ mm/d} \quad P_{eff} = 4.17 + 0.1P \text{ for, } P \geq 8.3 \text{ mm/d} \tag{5}$$

where P is daily rainfall.

The $ET_c$ (Equation (4)) is obtained from the multiplication of the reference crop evapotranspiration ($ET_o$) by a crop coefficient ($K_c$). In this study, the recommended Food and Agriculture Organization (FAO) Penman–Monteith equation is used [53,71]:

$$ET_o = \frac{0.408\Delta(R_n - G) + \gamma\frac{900}{T-273}u_2(e_s - e_a)}{\Delta + \gamma(1 + 0.34u_2)} \tag{6}$$

where $ET_o$ is reference evapotranspiration [mm day$^{-1}$], $R_n$ is net radiation at the crop surface [MJ m$^{-2}$ day$^{-1}$], G is soil heat flux density [MJ m$^{-2}$ day$^{-1}$], T is mean daily 2-m air temperature [°C], u2 is 2-m wind speed [m s$^{-1}$], $e_s$ is saturation vapor pressure [kP$_a$], $e_a$ is actual vapor pressure [kP$_a$], $e_s - e_a$ is saturation vapor pressure deficit [kP$_a$], $\Delta$ is slope vapor pressure curve [kPa °C$^{-1}$], and $\gamma$ is psychrometric constant [kPa °C$^{-1}$].

There is no radiation ($R_n$) parameter that can be directly compared, for both observation and the model dataset. Therefore, $R_n$ is derived from the Hargreaves radiation formula as a maximum and minimum temperature (Equations (7) and (8)). Since $G$ as the magnitude of the day or 10-day soil heat flux beneath the grass reference surface is relatively small, it may be ignored ($G_{day}$ = 0) [53].

$$R_n = (1 - alb)R_s - R_{nl} \tag{7}$$

$$R_s = k_{RS}\sqrt{(T_{max} - T_{min})}R_a \tag{8}$$

where alb is albedo or canopy reflection coefficient, which is 0.23 for the hypothetical grass reference crop [dimensionless], $R_s$ is the incoming solar radiation [MJ m$^{-2}$ day$^{-1}$], $R_{nl}$ is net outgoing longwave radiation [MJ m$^{-2}$ day$^{-1}$], $k_{Rs}$ is adjustment coefficient (0.16–0.19) [°C$^{-0.5}$], $T_{max}$ is maximum air temperature [°C], $T_{min}$ is minimum air temperature [°C], and $R_a$ is extraterrestrial radiation [MJ m$^{-2}$ d$^{-1}$]. For 'interior' locations, where landmass dominates and air masses are not strongly influenced by a large water body, $k_{Rs}$ = 0.16. In 'coastal' areas, situated over or adjacent to the coast of a large landmass and where air masses are influenced by a nearby water body, $k_{Rs}$ = 0.19 [53].

*3.4. Statistical Analysis*

To investigate the performance of each method (model and observation data comparison), this study uses several statistical evaluation tools. The correlation coefficient (R), mean absolute error (MAE), and bias were used to measure the performance of bias-correction methods.

$$R = \frac{\sum_{i=1}^{n}(O_i - \overline{O})(M_i - \overline{M})}{\sqrt{\sum_{i=1}^{n}(O_i - \overline{O})^2}\sqrt{\sum_{i=1}^{n}(M_i - \overline{M})^2}} \tag{9}$$

$$MAE = \frac{1}{n}\sum_{i=1}^{n}|M_i - O_i| \tag{10}$$

$$Bias = \frac{1}{n}\sum_{i=1}^{n}(M_i - O_i) \tag{11}$$

where n is the number of days *i*, while O and M are observation, and model data for each method applied, respectively. The correlation coefficient (R) measures the strength of linear relationship between two variables, The R closer to +1 indicate both variables have strong positive relationship and R closer −1 indicate both variables have strong negative relationship, while values closer to 0 indicate both variables have weak relationship.

Both MAE and bias measure the differences between two continuous variables. The bias describes systematic error of simulated dataset to under or overestimate from its reference values, while MAE measure magnitude of forecast errors. The MAE is the most reliable and useful measure widely used in model evaluations [72,73]. It has also been used in previous research [27,74].

## 4. Results

*4.1. Interrelation of Climate Variables*

The results of the inter-correlation of daily climate variables are summarized in Figure 2. Since most variables are not linearly correlated, Spearman rank correlation analysis is used to determine the relationship between climate variables [28]. Overall, a high positive correlation is shown by Pp, RH and temperature (Tx and Tn) for all datasets, while inter-correlation for other variables is relatively small. In terms of insignificant correlation, RAW has two insignificant correlation values for Pp-Tx and Pp-Ws. For UBC, only Pp-Tx is categorized as insignificant, whereas there is no insignificant value found for both observation and MBC datasets. The similarity between RAW and UBC inter-variable dependencies is shown in Figure 2b,c. In contrast to UBC, MBC inter-variables dependencies correspond closely to observations. The observation-MBC conformity can be clearly seen in the case of inter-correlation pattern of Pp-Tn and RH-Ws compared to observation-UBC (Figure 2a,b,d). This result agrees with previous studies [28,32], that UBC retains inter-variable dependencies as represented by RAW, while MBC interdependencies are closely related to the observation.

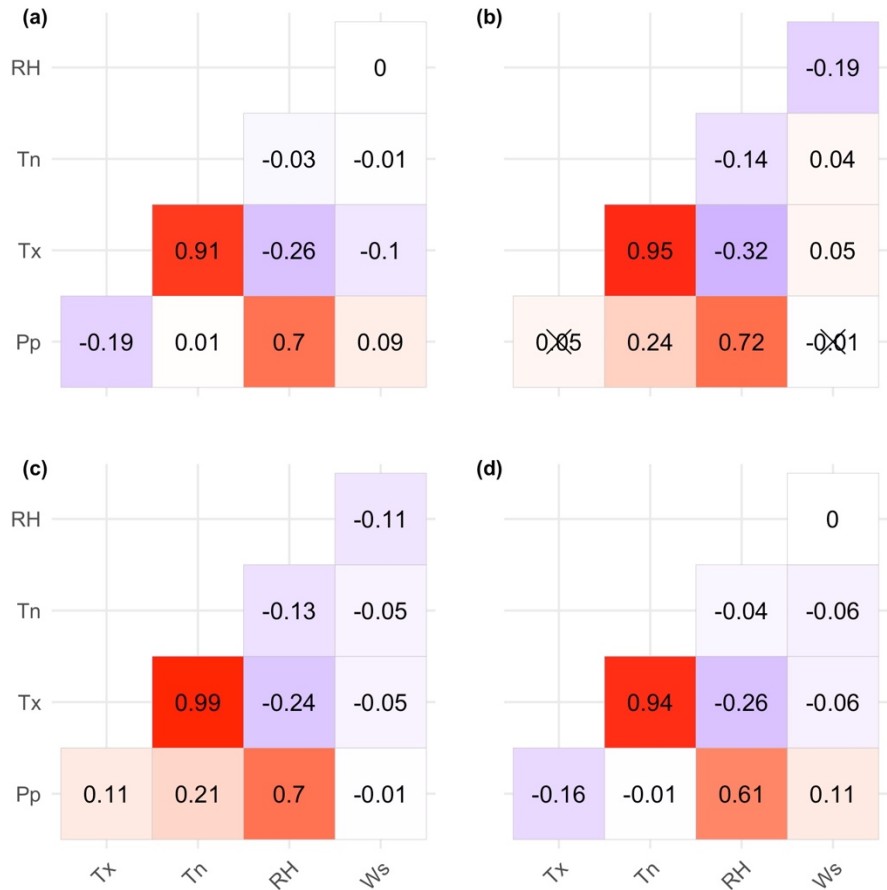

**Figure 2.** Inter-correlation of daily climate variables (precipitation (Pp), maximum temperature (Tx), minimum temperature (Tn), relative humidity (RH), and wind speed (Ws)) for (**a**) observation, (**b**) regional climate model data (RAW), (**c**) univariate bias correction (UBC), and (**d**) multivariate bias correction (MBC) for the 1989–2005 period. X denoted correlation values that are below significance level p = 0.05.

*4.2. Spatio-Temporal Bias Variability*

Figure 3 presents the daily seasonal mean bias of model (RAW), UBC and MBC, against observation data. The bias displayed in the boxplot explains the model deviation from its real value (observation). Generally, RAW presents overestimation (positive bias) of all variables, except for Tn which gives lower value (underestimation) than observation. From the figure above, all bias correction methods (UBC and MBC) successfully reduce RAW bias for all variables in all seasons. In terms of precipitation bias (Figure 3a), RAW gives slightly higher positive bias during March, April, May (MAM) and June, July, August (JJA) than during December, January, February (DJF) and September, October, November (SON), while both methods show improvement in minimizing those biases. Nevertheless, it should be noted that the large bias variability is present during the summer rainfall period (JJA) and both methods are unable to minimize the outliers. This is due to the distribution-based method used (such as QM and QDM), which is usually inadequate to represent the extreme tail of daily precipitation distribution.

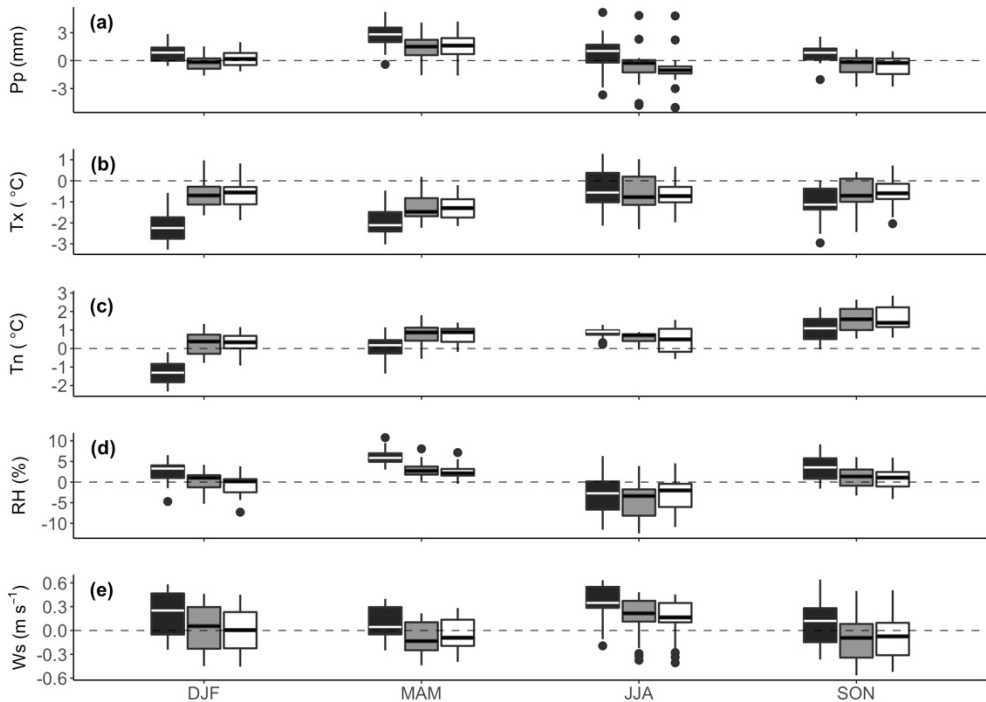

**Figure 3.** Boxplots of the daily seasonal mean bias of (**a**) precipitation, (**b**) maximum temperature, (**c**) minimum temperature, (**d**) relative humidity, and (**e**) windspeed of RAW (black), UBC (grey), and MBC (white) against ground-based observation for the 1989–2005 period.

The results of daily seasonal bias for Tx and Tn are shown in Figure 3b,c, respectively. Similarly, UBC and MBC bias-correction methods effectively reduce RAW Tx and Tn bias, especially for the winter season (DJF). On average, the MBC method is shown to have better performance to reduce variability for maximum temperature (indicated with smaller bar size). However, for all seasons, no significant predominance between the two methods are evident, while the MBC method appears to have slightly negative performance for Tn during SON period (Figure 3c). This may be the effect of MBC method which maintains inter-correlation of Tn toward Pp, RH, and Ws. Tn inter-correlation for observation and model data is significantly different compared to other variables (Figure 2).

The comparison between RAW and two bias-corrected methods for RH are presented in Figure 3d. Overall, the two bias-correction methods successfully reduce biases throughout the year, but MBC is consistently better than UBC. The JJA period seems to give large bias variability compared to other periods, while the UBC method give no improvement as compared to RAW during this period.

On the other hand, the Ws RAW tends to overestimate observation for all seasons (Figure 3e). The windspeed overestimation is supported by Li et al. [75], who compared Ws observation with CNMR-CM (Centre National de Recherches Météorologiques Climate Model National Center for Meteorological Research-Climate Model 5) model output. However, compared to other variables, Ws gives a wider bias variability during all seasons, while both bias-correction methods successfully reduces the biases, but bias variability still remains. Overall, MBC shows better performance than UBC in reducing Ws model bias.

In summary, both methods show good performance. However, the results are varied across season and variables, which makes it difficult to determine the best method overall. On the other hand, insufficient RCM performance during the JJA period has caused wider bias variability for all variables, especially Pp. This also affect bias correction methods which usually fail to resemble the extreme value during this period. Furthermore, this will also affect the simulation of IWNs conducted during the JJA–SON periods

Daily MAE value for each variable (Pp, Tx, Tn, RH, and Ws is presented on the heatmap plot (Figure 4) as a function of bias correction method (x-axis) and station name (y-axis)). For identification purposes, the MAE result is normalized using a min-max scaling feature which is relative to each variable. In general, both bias correction methods lead to good performance in reducing RAW bias. Both methods show large improvement on RH variable, while Pp and Ws present small adjustments. On the other hand, UBC correct both modeled temperatures (Tx and Tn) better than MBC. Furthermore, in terms of minimum temperature, the station averaged MAE (minus Lushan station) for RAW, UBC and MBC (2.3, 2.2 and 2.5, respectively) shows that MBC has no improvement in correcting RAW bias. This is also found in temporal bias (Figure 3), whereas MBC method attempt to maintains inter-correlation of Tn against other variables.

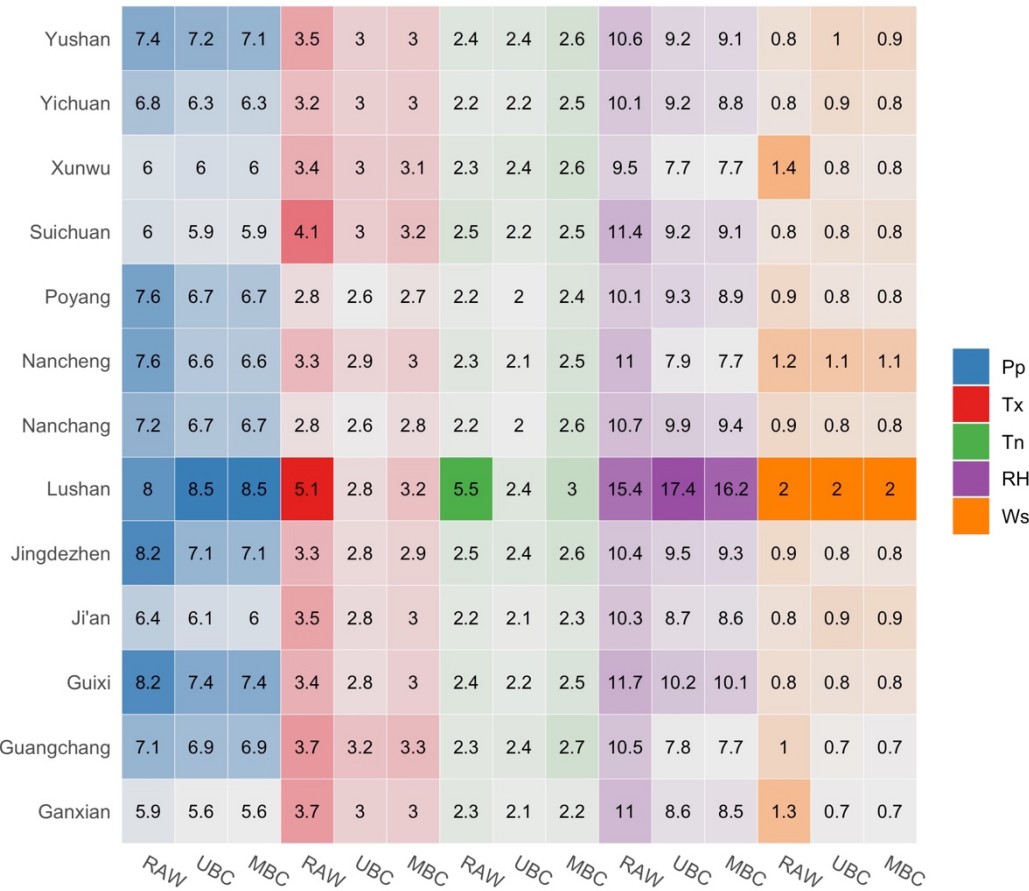

**Figure 4.** Heat plot of normalized daily mean absolute error (MAE) relatively for each variable (1989–2005). A lighter color denotes better result, with actual value overlaid within each cell.

As clearly seen, Lushan station has relatively high RAW bias compare to other stations, which is caused by model limitation to simulate orographic processes in high-elevation areas (Table 1). Similarly, Liu, Z. et al. [76] found large differences between observed and modeled data indicated by the difference between modeled (ERAINT) elevation and site elevation. However, temperature is relatively easy to bias-correct [48] compared to precipitation which is predicted through model parameterization of temperature and humidity from observation [77]. Furthermore, both UBC and MBC give no improvement for Pp, RH, and WS in areas with high topography (Lushan station). The result is consistent with previous research which gives less significance in precipitation bias correction for regions that experience significant orographic effects (such as mountainous areas) [27].

### 4.3. Rice Irrigation Water Needs (IWNs) Simulation

In this study, the calculation of rice IWNs is carried out only in the late rice period. This period begins on 1 June through 7 November (during the 2006–2007 simulation year), with the first 30 days being land preparation. The total duration of planting takes about 130 days from the initial stage to the late phase (harvesting). Figures 5 and 6 present the Q–Q plot of both potential evapotranspiration ($ET_o$) and effective rain during 2006–2007 rice IWNs simulation, consecutively. Both are the two main parameters forming IWNs derived from climate variables. The $ET_o$ is obtained from calculations involving Tx, Tn, RH, and Ws, while $P_{eff}$ is derived from the precipitation variable. Thus, the IWNs' accuracy depends on the bias correction skills for each climate variables.

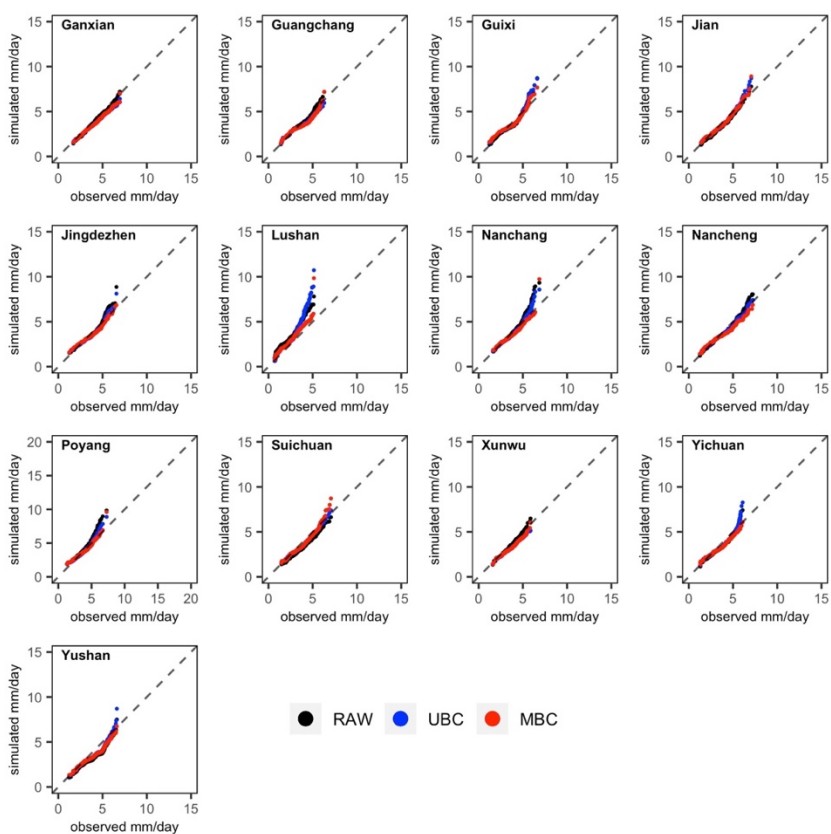

**Figure 5.** The Q–Q plot of daily potential evapotranspiration ($ET_o$) during late rice period (01 June–11 November).

In terms of $ET_o$ (Figure 5), generally in all cities, UBC and MBC have the good performance for correcting RAW bias which indicate that MBC and UBC closely followed the Observed $ET_o$ (dashed line). In a few cities UBC shows a strong overestimate with observed $ET_o$ greater than 5 mm/day (Guixi, Lushan, Nanchang, Poyang and Yichuan). The overestimation corrected better in MBC method, which appears to have better skill at correcting extreme values. Overall, the MBC method appears to have provided better performance than UBC. As can be seen from Table 3, on average, MBC correlates to observation better than UBC (R = 0.53 and 0.51, respectively), and provides smaller MAE value (MBC = 0.86 and UBC = 0.92).

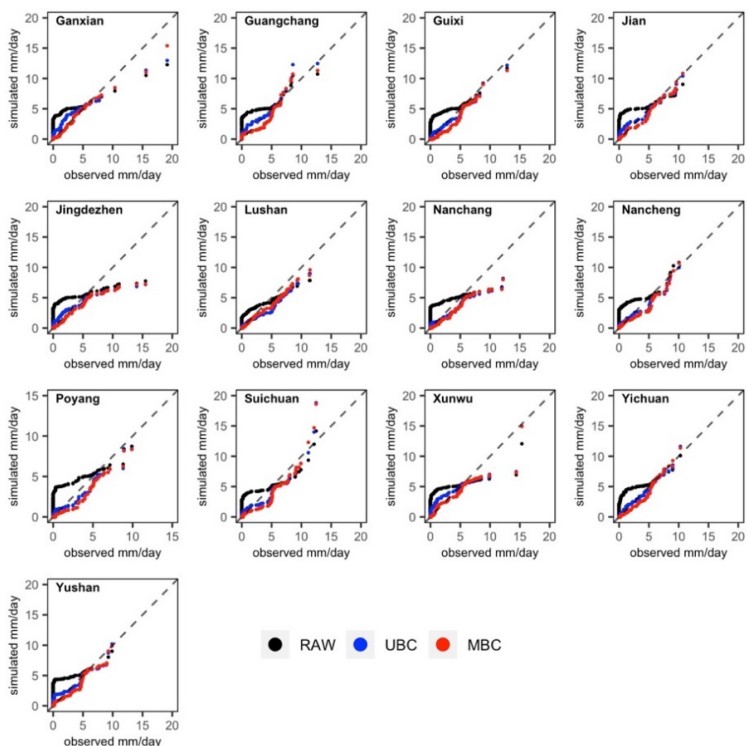

**Figure 6.** The Q–Q plot of daily effective rainfall ($P_{eff}$) during late rice period (01 June–11 November).

**Table 3.** The daily correlation coefficient and mean absolute error value for potential evapotranspiration and effective rain component of rice IWNs (R = RAW, U = UBC, and M = MBC against ground-based observation).

| Station | Potential Evapotranspiration (ET$_o$) | | | | | | Effective Rainfall (P$_{eff}$) | | | | | |
| --- | --- | --- | --- | --- | --- | --- | --- | --- | --- | --- | --- | --- |
| | Corr | | | MAE | | | Corr | | | MAE | | |
| | R | U | M | R | U | M | R | U | M | R | U | M |
| Ganxian | 0.40 | 0.50 | **0.51** | 1.07 | **0.94** | **0.94** | **0.17** | 0.13 | 0.12 | 2.15 | 1.86 | **1.77** |
| Guangchang | 0.49 | **0.56** | **0.56** | 0.92 | **0.80** | 0.81 | 0.23 | **0.24** | 0.22 | 2.19 | 1.66 | **1.57** |
| Guixi | 0.58 | 0.58 | **0.59** | 0.89 | 0.87 | **0.81** | 0.33 | 0.33 | **0.34** | 2.13 | 1.65 | **1.55** |
| Jian | **0.55** | 0.54 | **0.55** | 0.94 | 0.92 | **0.88** | **0.27** | 0.26 | 0.21 | 2.12 | **1.57** | 1.58 |
| Jingdezhen | 0.53 | **0.56** | **0.56** | 0.90 | 0.82 | **0.79** | **0.25** | 0.20 | 0.15 | 2.10 | 1.62 | **1.58** |
| Lushan | **0.35** | 0.29 | 0.32 | 1.23 | 1.36 | **1.01** | **0.10** | 0.08 | 0.08 | 2.23 | **2.03** | 2.09 |
| Nanchang | 0.47 | 0.49 | **0.52** | 1.01 | 0.92 | **0.80** | 0.27 | 0.30 | 0.31 | 1.94 | 1.38 | **1.29** |
| Nancheng | 0.56 | 0.57 | **0.59** | 0.95 | 0.91 | **0.89** | 0.26 | 0.31 | 0.33 | 2.18 | 1.60 | **1.50** |
| Poyang | 0.44 | 0.47 | **0.48** | 1.06 | 0.90 | **0.82** | 0.26 | **0.27** | 0.27 | 1.83 | 1.25 | **1.19** |
| Suichuan | 0.54 | **0.55** | 0.54 | 1.02 | **0.91** | 0.94 | **0.19** | 0.13 | 0.10 | 2.33 | 2.10 | **2.07** |
| Xunwu | 0.27 | **0.39** | **0.39** | 0.87 | 0.79 | **0.78** | 0.20 | **0.22** | 0.21 | 2.47 | 2.15 | **2.09** |
| Yichuan | 0.57 | 0.57 | **0.60** | 0.90 | 0.89 | **0.81** | **0.22** | 0.21 | 0.20 | 2.20 | **1.74** | **1.74** |
| Yushan | 0.61 | 0.59 | **0.61** | 1.00 | 0.93 | **0.90** | 0.34 | **0.37** | **0.37** | 2.22 | 1.68 | **1.58** |
| Mean | 0.49 | 0.51 | **0.53** | 0.98 | 0.92 | **0.86** | **0.24** | 0.23 | 0.22 | 2.16 | 1.71 | **1.66** |

\* Bold value denoted better result.

On the other hand, in general, the RAW shows a strong overestimating trend in lower $P_{eff}$ values (0–5 mm) and underestimates in high $P_{eff}$ values for all stations (Figure 6). Furthermore, result shows that the performance of both methods show better skill in correcting RAW at low values, but less successful in correcting extreme values. In detail, it can be seen from the data in Table 3 that MBC provides a better bias correction (smaller MAE) than UBC except for Jian (UBC = 1.57, MBC = 1.58) and Lushan (UBC = 2.03, MBC = 2.09). Overall, MBC shows better improvement in daily station average correlation and MAE than UBC.

Figure 7 shows the simulation of rice daily IWNs during the 2006-2007 late period in Jiangxi Province. The calculation of the components of rice IWNs is conducted daily but displayed on a monthly graph to facilitate interpretation. The rice IWNs are shown accumulated for a whole month, except for November (accumulated for seven days).

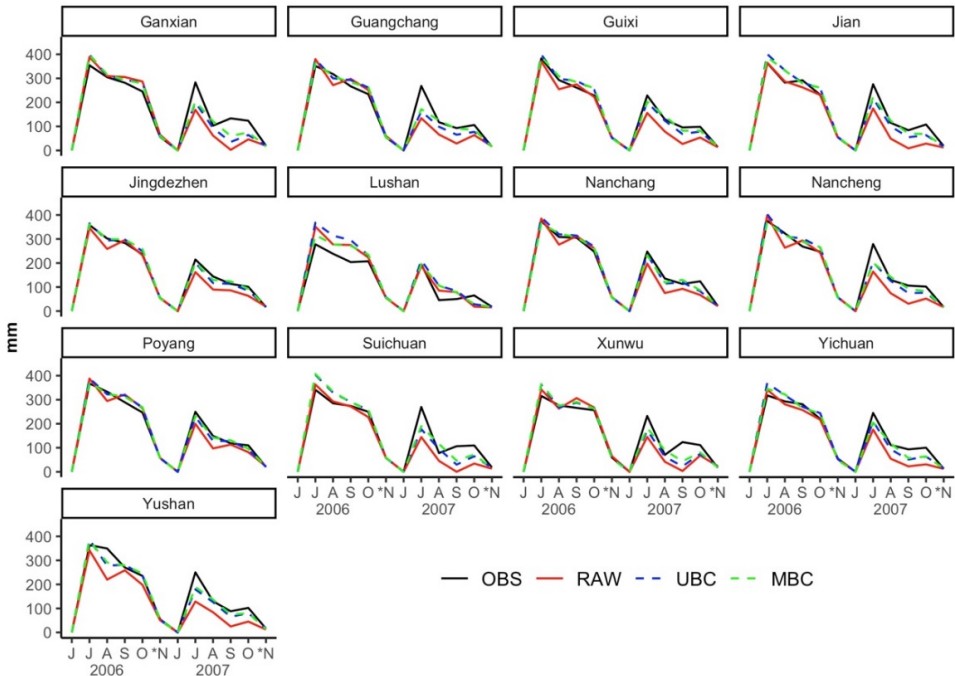

**Figure 7.** Daily (monthly aggregated) irrigation water needs (IWNs) for rice late period in Jiangxi Province during the 2006–2007 simulation period (*N only seven days simulation).

Generally, the RAW simulation IWNs presents underestimation for all stations compared to observation with an exception for Lushan, which gives a higher value for RAW. The Lushan IWNs overestimation caused by the poor climate model performance (RAW). Furthermore, the average rice IWNs in the 2006 growing season was higher than in 2007. According to Liu et al. [56], the average potential evapotranspiration for Jiangxi province in 2006 was more significant than in 2007, which was one of the causes of differences in the value of rice IWNs.

Overall, the two correction-bias methods used can correct the bias that exists between observation and the model. Moreover, in 2007, observation and RAW difference is higher compared to the previous year, but both UBC and MBC successfully reduce the existing bias. Similar to the last analysis (Figure 4), an unrealistic result is obtained for the Lushan area (overestimation) due to UBC and MBC's inability to correct the existing bias, especially for precipitation.

Table 3 shows the summary of statistic evaluation for the IWNs component ($ET_o$ and $P_{eff}$) to provide a detailed performance of each bias correction method correlation and bias. Overall, both bias-correction method improved RAW $ET_o$ and $P_{eff}$. In detail, as expected, both methods improved $ET_o$ values better than $P_{eff}$. This improvement caused by $ET_o$ calculated using well bias-corrected climate variables (such as Tx, Tn, RH, and Ws).

In general, MBC showed an outstanding performance compared with UBC for both IWNs' components. Furthermore, in the context of the correlation coefficient for $ET_o$, both methods improve RAW correlation, while MBC has a better correlation value than UBC (R = 0.53 and R= 0.51, respectively). Similarly, the improvement was also shown by MBC and UBC for bias reduction with MAE values of 0.86 and 0.92, respectively. This indicates the MBC correct model bias is better than UBC. In terms of the $P_{eff}$ component, positive result also shown by MBC and UBC for bias correction. The average MAE value for MBC is smaller than UBC (MAE = 1.66 and MAE = 1.71, respectively), this means that

the MBC corrects the model bias better than UBC. The correlation coefficient for the effective rainfall component did not show any significant result for both bias correction methods. As shown in Table 3, the average correlation coefficient for RAW, UBC, and MBC is 0.24, 0.23, and 0.22, respectively.

## 5. Conclusions

RCM simulations are generally subjected to substantial systematic biases as compared to observations, making the outputs unsuitable for assessing agricultural-climate impact directly. This study systematically tests the effects of univariate bias correction methods such QM versus a multivariate bias correction. The climate variables (Pp, Tx, Tn, RH, and Ws) are derived from the raw outputs of HadGEM3-RA forced with ERA-Interim reanalysis (referred to as RAW). Those meteorological datasets are fed to rice the IWNs equation for 13 ground-based observations in Jiangxi Province. The control period is set to 1989–2005, and 2006–2007 was a validation period.

The RAW model bias is dependent on the time, place, and variables analyzed. The RAW generally has a higher average annual value than the observation (overestimate) except for the maximum temperature (underestimate) during the control period 1989–2005. Moreover, high-topography areas such as Lushan (1164.5 m) tend to have higher biases than other regions with lower elevations. While based on the variables tested, relative humidity has the most significant average difference compared to other climate variables shown by the MAE value. This study also shows that UBC mostly retains inter-variable dependencies which may affects the reliable assessment of climate change impacts.

Both bias-correction methods successfully improved the simulation of late rice IWNs in Jiangxi province for the 2006–2007 period. The MBC performs better than UBC for correcting bias and improving correlation of $ET_o$ component, while for $P_{eff}$, both methods insignificantly improved the correlation coefficient, although MBC still has a better performance than UBC for $P_{eff}$ bias correction. In detail, both methods successfully reduced $ET_o$ and $P_{eff}$ RAW bias at low values but showed weak performance in correcting extreme bias.

These methods improved both RAW model bias and correlation. However, both approaches have low bias-correction performance on precipitation compared to other variables, as well as low performance in high-altitude areas. Another issue was noted during the calculation of late rice IWNs during the summer (rice-growing season), where the performance of the RAW model in summer was lower than the performance in winter. These three elements will significantly affect the calculation of rice IWNs.

**Author Contributions:** Conceptualization, W.H.; Methodology, W.H., J.Y. and J.Z.; Software, W.H. and L.C.; Validation, W.H. and J.Y.; Formal Analysis, W.H. and J.Z.; Investigation, W.H., J.Y. and J.Z.; Resources, W.H., J.Y., J.Z. and L.C.; Data Curation, L.C.; Writing—Original Draft Preparation, W.H.; Writing—Review and Editing, W.H., J.Y. and J.Z.; Visualization, W.H., J.Y. and L.C.; Supervision, J.Y.; Funding Acquisition, W.H., J.Y., J.Z., and L.C. All authors have read and agreed to the published version of the manuscript.

**Funding:** This research was funded by The National Natural Science Foundation of China (No. 41575111 and 41175098).

**Acknowledgments:** We thank the China Meteorological Administration and East Asia Coordinated Regional Climate Downscaling (CORDEX-EA) phase I, under the World Climate Research Program (WCRP) for providing the data for this research. The authors also are grateful to the anonymous reviewers for their constructive reviews that improved the quality of this study.

**Conflicts of Interest:** The authors declare no conflict of interest.

## Abrreviations

GCMs = Global Climate ModelsRCMs = Regional Climate ModelsECMWF = European Centre for Medium-Range Weather ForecastsERAINT = ECMWF Re-Analysis-Interim CORDEX-EA = Coordinated Regional Climate Downscaling Experiment-East AsiaNIMS = National Institute of Meteorological SciencesUBC = Univariate Bias CorrectionMBC = Multivariate Bias CorrectionIWNs = Irrigation Water NeedsCMA = China Meteorological AdministrationHadGEM3 = Hadley Centre Global Environment Model version 3MAE = Mean Absolute ErrorQM = Quantile MappingQDM = Quantile Delta MappingCNMR-CM5 = Centre National de Recherches Météorologiques Climate Model National Center for Meteorological Research-Climate Model 5CDF = Cumulative Density FunctionWCRP = World Climate Research ProgramPDF = Probability Density Function

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
