# Peer review of "Impact Analysis of Univariate and Multivariate Bias Correction on Rice Irrigation Water Needs in Jiangxi Province, China"

_water, doi:10.3390/w12020381_

Round 1
Reviewer 1 Report
In this manuscript, the authors applied some bias corrections methods to improve the prediction of the rice irrigation water nNeeds variables The paper is clear, well written and provides useful information for regional applications. However, there are some issues the authors should address before the publication.
First, it should be explained in the introduction what is expected to happen due to climate change and what sources of projections have been consulted for the study
Second, a picture of the region studied with the marked stations would help to locate it.
Third, the explanation of multivariate corrections should be further developed.
fourth, figure 4 should be slightly larger or change the scale to make it easier to see.
finally add a section of abbreviations at the end.
Reviewer 2 Report
In general, I think research on this subject can be relevant if conducted and presented correctly. But the provided manuscript lacks behind in both a proper introduction, in the implemented methods, interpretation of results and in the presentation of the research findings. The language (grammar) is very poor.
Introduction and background on bias corrections is very short. The introduction should provide the required background information, study question, hypothesis, and general approach. If the Introduction is done well, there should be no question in the reader’s mind why and on what basis you have posed a specific hypothesis. This, however, is not the case in the provided manuscript. In fact, it does not include any research hypothesis!
There is no justification as to why different bias correction methods were tested. Aren't there already enough studies out there? In what way is your study different? What insights can the scientific community gain from these results?
In terms of methods, there are also considerable problems:
First of all, if you plan to provide a comparison, why were only so few bias correction methods chosen?
In addition, there is data for only 18 years. When studying climate and climate models, one should consider longer periods (e.g. climate normals of 30yrs). Especially for a split sample test, 18 years is not enough. And evaluating the performance over 2 years is not reliable at all. When looking at 2 years, we can barely make any conclusions about climate change signals.
The MBC method is not at all explained in the manuscript. It is not okay to simply use a "black-box" R script without knowing what is going on inside. Instead, a formal mathematical description of MBC with advantages and disadvantages should be provided.
In the results, I didn't understand why there was such a long section dedicated to the comparison of observations and ERAINT. Why is this comparison important? It was not mentioned in the intro or methods that you would focus on this comparison. Also, the different bias correction methods were compared to observations alongside ERAINT data. You never really explained why ERAINT is of any interest in this comparison. This clearly shows that there is insufficient description in the methods. E.g., were the bias-correction methods calibrated on ERAINT data or observations?
By reading the manuscript, I didn't get convinced that the authors are familiar with the proper scientific language and literature in the field. The terms 'parameters' and 'variables' are mixed up in the text and references are used wrongly (e.g., e.g. page 4, line 125: the split sample test for bias correction methods was originally proposed by Teutschbein & Seibert 2013 doi:10.5194/hess-17-5061-201, and not Sitanek 2017).
In general, the authors simply stated that the tested methods are different, but made no attempt to understand why they perform differently. The manuscript completely lacks a discussion section where the results are properly interpreted and discussed in a wider context.
Therefore, I do not think that this manuscript is of sufficient quality to be further considered for publication. Substantial revisions in the methodology as well as in the writing of the manuscript are required.
